# Experimental Setup for Evaluating Depth Sensors in Augmented Reality Technologies Used in Medical Devices

**DOI:** 10.3390/s24123916

**Published:** 2024-06-17

**Authors:** Valentyn Stadnytskyi, Bahaa Ghammraoui

**Affiliations:** Center for Devices and Radiological Health, U.S. Food and Drug Administration, 10903 New Hampshire Avenue, Silver Spring, MD 20993, USA

**Keywords:** augmented reality, virtual reality, validation, evaluation, image registration

## Abstract

This paper presents a fully automated experimental setup tailored for evaluating the effectiveness of augmented and virtual reality technologies in healthcare settings for regulatory purposes, with a focus on the characterization of depth sensors. The setup is constructed as a modular benchtop platform that enables quantitative analysis of depth cameras essential for extended reality technologies in a controlled environment. We detail a design concept and considerations for an experimental configuration aimed at simulating realistic scenarios for head-mounted displays. The system includes an observation platform equipped with a three-degree-of-freedom motorized system and a test object stage. To accurately replicate real-world scenarios, we utilized an array of sensors, including commonly available range-sensing cameras and commercial augmented reality headsets, notably the Intel RealSense L515 LiDAR camera, integrated into the motion control system. The paper elaborates on the system architecture and the automated data collection process. We discuss several evaluation studies performed with this setup, examining factors such as spatial resolution, Z-accuracy, and pixel-to-pixel correlation. These studies provide valuable insights into the precision and reliability of these technologies in simulated healthcare environments.

## 1. Introduction

Extended Reality (XR) technologies emerged in the 1970s and were first commercialized in the 1990s [1]. Recent advances in computer chips, machine vision, and algorithms have made XR headsets portable, facilitating their use in medical settings. In the medical field, the application of XR is referred to as Medical Extended Reality (MXR). This term encompasses a range of technologies and applications. Head-mounted displays (HMDs) are commonly used in MXR, with virtual reality (VR) HMDs presenting virtual objects in a virtual world; while augmented reality (AR) HMDs overlay virtual objects onto the real environment, augmenting reality with virtual content.

Although the AR development has been rapid, it faces significant challenges in healthcare applications. The seamless visualization of and interaction with digital content in the real world heavily relies on multiple HMD built-in sensors, e.g., depth cameras, inertial motion units, and color and monochromatic cameras for environmental sensing. For instance, depth cameras are used to enhance and enable intuitive gesture recognition [2,3,4], facilitating realistic virtual object placement and interaction with real-world elements. In summary, depth cameras play a fundamental role in enabling immersive AR experiences by providing accurate spatial understanding and facilitating natural interaction with virtual content. The challenges include integrating AR technology seamlessly into clinical workflows, ensuring patient safety and privacy, and overcoming technical limitations such as accuracy and field of view [5,6]. Despite these challenges, a growing number of AR applications have been used in training, education, telemedicine, preoperative planning and postoperative assessment, and intraoperative systems [7,8]. For instance, AR HMDs enable capturing of images and videos from a surgeon’s perspective and share them with remote users [9]. Novel clinical applications are also emerging, providing engaging environments and training for physiotherapy [10] and pain management [11,12], as well as the gamification of physical therapy.

Amid this rapid development and widespread adoption of AR technologies across various applications, evaluating these technologies to assess their reliability has received comparatively less attention. Most efforts have been directed toward evaluating these applications as a whole using qualitative methods [13,14,15,16], which rely heavily on subjective user feedback and focus on whether a very specific task can be completed better or worse using applications reliant on depth sensors. For example, the review paper discusses various evaluations of gesture tracking enabled by depth cameras [17], and a different collection of works examines improvements in the performance of simultaneous localization and mapping (SLAM) algorithms when depth cameras are used [18,19]. However, these studies primarily focus on evaluating the performance of algorithms or the enhancement of task performance, rather than directly assessing the performance of the depth cameras themselves. This predominant reliance on subjective metrics underscores the need for an alternative approach emphasizing quantitative evaluation and validation of AR technologies in healthcare settings and focusing on the performance of individual sensors and their contributions to the application’s performance as a whole.

In this work, we have developed a fully automated benchtop experimental test setup designed for quantitative evaluation of augmented reality sensors. In this paper, we aim to assess the performance of depth cameras within AR headsets under conditions closely resembling real-world scenarios, e.g., rotation and translation of user’s head. This paper initially describes the system architecture and the process of automated data collection. Subsequently, we present several evaluation studies conducted using the Intel RealSense L515 LiDAR camera as an example.

## 2. Materials and Methods

### 2.1. Experimental Design

The evaluation experimental setup, illustrated in Figure 1a, was designed to create a controlled and stable testing environment for evaluating range sensing technologies, with a specific focus on depth camera sensors and augmented reality. This setup aimed to assess the performance of these technologies under various conditions, including motion, position and against different test objects.

The experimental setup, mounted on an anti-vibration optical bench table measuring 6 × 3 feet^2^ (model INT3-36-8-A Integrity 3 VCS by Newport Corp., Irvine, CA, USA), is organized into two primary sections. Its left side featured the scanning hardware section, while the remaining space was dedicated to the test object mounting area, as shown in Figure 1a.

To achieve precise and controlled movements, we employed a motion control system from Zaber Technologies(Vancouver, BC, Canada), configured in a gantry system. Stages were stacked on top of each other. The lowest $X_{m}$-axis of this gantry comprised two actively driven translational stages operating synchronously in a lock-step mode, while the middle $Y_{m}$-axis linear stage was mounted atop the two $X_{m}$-axis linear stages. A rotational stage was positioned on top of the $Y_{m}$-axis linear stage. The $X_{m}$ and $Y_{m}$ axes translational stages, featuring meter-long linear guides (LC40B10000) equipped with stepper motors and encoders (NEMA23), provided an accuracy of 60 μm and repeatability of less than 20 μm across their entire range. Of note, the custom software improvements were made to achieve performance specifications higher than the standard ones posted on the vendor’s webpage. The improved accuracy and repeatability were verified using laser interferometry. The motorized rotational stage is capable of remarkable precision, offering a repeatability of less than 0.02 degrees and an accuracy of 0.08 degrees without additional modifications. These specifications are well suited for evaluating standalone depth camera and built-in HMD sensors and are well beyond the typical depth resolution (on the order of 250 μm) of current depth cameras on the market.

To enhance modularity and flexibility, all devices under test (DUT) are mounted on separate platforms. The preconfigured platforms could be swapped in and out in under five minutes. Each of the platforms were equipped with Ethernet, USB 3.0, and DC power connections, providing a total power delivery of 150 W and several different modes of connectivity, supporting a variety of USB devices, Ethernet devices and portable computing systems such as Raspberry Pi or NVidia Jetson Nano GPU systems. We utilized solid aluminum breadboards (12 inches × 12 inches × 0.25 inches) from Base Lab Tools, USA, as our mounting platforms. Post-processing machining of the platforms aligned the central four holes arranged on a one-inch grid with four 25 mm-spaced mounting holes on the rotational stage, enabling the support of objects weighing up to 50 kg. These platforms served as a means to assemble and prepare devices under evaluation and for testing necessary software before mounting them on the rotation stage, thus facilitating independent work without requiring direct access to the evaluation experimental setup.

Each platform, featuring the preconfigured DUTs, was mounted on a rotation stage above $X_{m}$–$Y_{m}$ translation stages. This configuration offered three motorized independent movements: two translational and one rotational. This arrangement enabled for the evaluation of range sensing by DUTs with three degrees of freedom. Similarly, the cameras mounted on the platform could have an extra degree of freedom achieved via manual or motorized tilt stage, four total degrees of freedom, including rotation around a vertical axis and tilting around a horizontal axis, as depicted in Figure 1c. All stages in our setup were capable of controlled movements at varying speeds and accelerations. This flexibility was essential for assessing camera responses during motion and at different locations, allowing us to simulate a wide range of user medical related activities and healthcare scenarios.

While our system’s current configuration could support various depth cameras and a HoloLens 2 augmented reality headset, as indicated in Figure 1b,c this study focused on presenting results using the Intel RealSense L515 LiDAR camera as an example.

In addition to the hardware components, we developed a series of test objects. These were intended to simulate real-world scenarios and play a crucial role in our testing procedures, a sample of which is shown in Figure 2. Figure 2a features a slanted edge and the motorized screen mounted on the optical table in the test object observation area. The slanted edge is positioned in front of the screen (TB4 - Black Hardboard, 24 × 24 inches), which acts as an adjustable background that can be positioned at various distances away from the slanted edge.

The second is a flat wall test object and shown in Figure 2b, and a large flat area of drywall located outside of the test object positioning area due to its size.

Our experimental setup with integrated components and flexible configurations was intended to provide a dependable and flexible testing environment for evaluating depth cameras and other range sensing technologies.

### 2.2. Modular Experimental Setup with Integrated Control and Data Collection System

The experimental setup was designed with a modular approach, enabling quick and seamless alternation of devices under investigation. These devices could be swiftly mounted on top of the rotational stage without impacting the overall system functionality. We utilized a Raspberry Pi 4 Model B Quad Core 64 Bit WiFi Bluetooth (8 GB) (Raspberry Pi Foundation, Cambridge, United Kingdom) for synchronizing the various components. Our setup included three different depth cameras connected to the same Raspberry Pi. The Raspberry Pi was networked via an Ethernet cable and ran on an Ubuntu 20.0 Linux system. This setup supports server applications and facilitates high-speed data transfer, with Ethernet bandwidths up to 10 Gbps.

In Figure 1b, we show a platform with an Intel Real Sense L515 LiDAR depth camera mounted on it. This camera is powered through and is connected to the control computer via a USB 3.0 Power Delivery (PD) port on the USB hub mounted on the platform. The control computer uses Python-based software version 3.9 that establishes connection with, controls and retrieves data from the camera.

Alternatively, as depicted in Figure 1c, a different platform may accommodate the HoloLens 2 augmented reality headset. The HoloLens 2 is powered through a USB 3.0 Power Delivery (PD) port and connects to the control computer directly via WiFi. Connectivity with the HoloLens 2 is established through a server application, written in C programming language, running on the HoloLens 2 AR device. A Python-based client process retrieves real-time data from the built-in sensors of HoloLens 2.

The data collected were frames that reflect the depth measurements captured by the cameras. These frames can be saved in various formats for analysis. Users can configure the motion trajectories, speed, and acceleration of the test objects and cameras during data collection using a pre-set dataset of points, which served as input for our data acquisition software; while the system allows for dynamic positioning, in this study, we focused on examples of evaluation under static conditions.

#### Systematic Calibration and Alignment Process for Depth Cameras

This calibration procedure consisted of two primary phases: intrinsic calibration and extrinsic alignment.

In the first stage, we executed intrinsic calibration in strict accordance with the manufacturer’s guidelines, as referenced in [20]. This critical step was designed to rectify lens distortions and mitigate any sensor misalignment within the camera. It also encompassed inter-camera alignment, in which we employed a combination of advanced software algorithms and precise manual adjustments to ensure a synchronized and highly precise depth perception across multiple sensors. For an in-depth understanding of our bench-top calibration method, we encourage readers to explore the comprehensive details provided in a published paper [21].

After intrinsic calibration, we moved to the extrinsic alignment phase, a crucial step that aligns the captured depth data with real-world coordinates custom-tailored to match our study’s specific requirements. For instance, when aligning a depth camera with a flat wall test object, our approach utilized root-mean-square error minimization within two regions of interest (ROI): the horizontal and vertical dimensions. In the horizontal ROI, we meticulously selected a set of five rows of pixels, symmetrically centered around the central row of pixel in the depth image. By vertically binning these pixels, we enhanced the signal-to-noise ratio by a factor of $\sqrt{5}$ and derived a vector of distances along a horizontal axis. These central rows of pixels represent distances across the image and should be the same if the camera’s optical axis is perpendicular to the wall. This resulting vector is then subjected to a straight-line fitting process, enabling us to extract the yaw angle. The minimization of this angle plays a pivotal role in achieving a high-precision alignment of the camera with the test object.

We repeated this measurements for the vertical ROI, affording us real-time visualization of the pitch angle. By minimizing the pitch angle, we ensured a robust and accurate alignment of the camera with the test object, further enhancing the reliability of our measurements.

In summary, our benchtop evaluation setup employed a systematic calibration procedure that adhered closely to manufacturer specifications while incorporating our in-house developed alignment techniques. This thorough calibration process ensured that our depth cameras performed optimally and delivered precise and synchronized depth perception across various sensors.

## 3. Examples of Evaluation Studies

The depth cameras return an array of data with three coordinates X, Y, Z, where Z is the depth coordinate, and X and Y are in object plane which is perpendicular to the depth axis. The examples of evaluation metrics in following section aim at establishing accuracy and precision of numbers reported by depth cameras.

Section 3.1 measures the modulation transfer function (MTF) of a sharp change in Z values created by a physical slanted edge.

Section 3.2 provides an example of evaluation metric that looks at precision and accuracy of Z measurements for fixed X and Y coordinates.

Section 3.3 examines the correlation of temporal noise profiles between different sets of pixels which revealed long range correlation.

### 3.1. Image Plane Spatial Resolution

In this section, we explore how the spatial resolution of a depth camera varies with different contrast levels. Here, "contrast" is defined as the range of depth values detected by the camera. Our method involved using a slanted edge test object (Figure 2a) positioned at a constant distance from a uniform background panel, creating a distinct difference in depth values between the test object and the panel. The slanted edge is commonly used to determine modulation transfer function (MTF) in optical systems and allows us to determine a system’s ’sharpness’ performance at all spatial frequencies in one go [22]. Images of this arrangement were captured with the back panel set at various distances 10 mm, 50 mm, and 160 mm—as illustrated in Figure 3. We then performed 2D Modulation Transfer Function (MTF) measurements on each image to assess the spatial resolution within the image plane, utilizing an open-source software tool for standard 2D MTF analysis [23].

The resulting MTF data, as shown in Figure 4, were acquired using the Intel RealSense L515 LiDAR range sensing camera. Interestingly, we observed a significant reduction in resolution with decreasing contrast. This finding highlights that spatial resolution deteriorates as the distance between objects in the field of view lessens, indicating a dependency of the camera’s resolution in the $X_{c}$–$Y_{c}$ plane on the contrast in image depth values, with *c* representing the camera.

### 3.2. Z-Precision and Z-Accuracy Measurements

We aimed to assess the precision and accuracy of distance measurements obtained from depth cameras, employing specific metrics to provide a comprehensive evaluation. Assessment of precision in distance measurements serves as a crucial metric, enabling the placement of error bars on the distance values returned by depth cameras. In this method, the depth camera was first positioned at a fixed distance of $z_{j=0}^{truth}$ from a flat-wall test object. We performed careful configuration and alignment to ensure that the optical axis ($Z_{c}$-axis) of the camera was perpendicular to the test object and aligned with the $Y_{m}$-axis of the gantry system. Next, we precisely repositioned the depth camera under evaluation using an automated motion control system at M different distances $z_{j=1..M}^{truth}$ from the flat wall test object. At each of these positions, we collected and saved a dataset comprised of *N* depth images. The acquired data were analyzed and $\bar{z_{j}}$—mean value, $RMSE_{j}$—root-mean-square error, $z_{j}^{acc}$—accuracy in z at *j* positing, and $\sigma_{j}$—standard deviation were computed for each position ($z_{j}^{truth}$) of the camera.

Root-mean-square error (RMSE) between the observed distance $z_{ij}$, obtained using a depth sensing camera and the known values $z_{j}^{truth}$ of depth camera position, were calculated using a single pixel in the center of the depth image. The RMSE is calculated as follows:


(1)
$$RMSE_{j}=\sqrt{\frac{\sum_{i=1}^{N}{\left(z_{ij}-z_{j}^{truth}\right)}^{2}}{N}}$$


Here, *i*—represents image index, *j*—represents positing index, *N* represents the total number of images collected at fixed position of the depth camera with respect to the flat wall test object.

Accuracy of the z-value measurement, as measured by the depth camera as a deviation of measured mean values $\bar{z}$ and the known camera position, the ground truth. The mean value can be calculated as follows:


(2)
$$\bar{z_{j}}=\frac{\sum_{i=1}^{N}z_{ij}}{N}$$


Here, $z_{i}$ represents the values of depth returned by the depth camera for a given pixel. Accuracy in z—$z_{acc}$ can be calculated as follows:(3)$$z_{j}^{acc}=z_{j}^{truth}-\bar{z_{j}}$$

Precision of the z-value measurement, as measured by the depth camera ($\sigma_{z}$) of measured depth values across different positions of depth camera with respect to the flat wall test object, is calculated as follows:


(4)
$$\sigma_{j}=\sqrt{\frac{\sum_{i=1}^{N}{\left(z_{ij}-\bar{z_{j}}\right)}^{2}}{N}}$$


As an example, we positioned the camera at $z_{0}^{truth}$ = 500 mm away from the flat wall. We confirmed the distance using the external ruler and collected 1000 images to ensure good statistical accuracy in measuring of $z_{i}$ and in establishing one-to-one correspondence between $Y_{m}$ encoder readings and physical distance. Later, we repositioned the depth camera to $z_{i}^{truth}$ = 600 mm, 700 mm, 800 mm, 900 mm and 1000 mm, collected 100 images (N = 100) from each location, and computed the metrics mentioned above.

Figure 5a displays the measured distances by the central pixel for all six locations, offering insights into how distance measurements vary with camera placement. In Figure 5b, for each position, we present the standard deviation and RMSE of the measured position. Notably, the standard deviation exhibits degradation as the camera is positioned farther away from the test object and RMSE drastically degrades due to large errors in measured mean position and actual position. Figure 5c illustrates the difference between the measured distance and the position specified by precise translational stages, read from built-in encoders. This comparison highlights errors in Z-accuracy, emphasizing that calibration at a single anchor point is insufficient for ensuring accurate calibration across the entire range of distances.

Finally, Figure 5d Panel D showcases six histograms of independent measurements obtained from 100 sequential depth images acquired at six known distances ($z_{j}^{truth}$). These values were fitted to a Gaussian function, enabling us to extract mean and standard deviation values from the fit. This analysis provides valuable insights into the distribution of depth measurements at different distances, as well as accuracy and precision of distance measurements.

Through this evaluation of precision in distance measurements, our setup and analysis shed light on the variations and accuracy considerations associated with depth camera performance across varying distances.

### 3.3. Pearson Pixel-to-Pixel Correlation

Within the framework of the setup detailed in this paper, one can efficiently compute correlations between distance measurements obtained from distinct pairs of pixels. This correlation analysis employs the Pearson correlation coefficient [24,25], providing valuable insights into the relationships between these measurements.

The dataset under examination comprises 1800 sequentially measured images, equivalent to 60 s of data collection at a frame rate of 30 frames per second. In Figure 6a, Panel A, we present static distance measurements obtained from a camera positioned facing a flat wall. Specifically, three traces are displayed: one for the central pixel, another for a pixel situated 100 pixels above the center, and a third for a pixel located 100 pixels to the right. To provide a clear overview of how average depth measurements evolve over time, we incorporate a solid line to represent the averaged data. Notably, these three traces exhibit significant correlation, particularly at larger distances of approximately 100 mm within the XY plane.

In Figure 6b, we show the correlation map generated by correlating the central pixel with all other pixels within the dataset, enabling us to explore comprehensive patterns of correlation.

Figure 6c displays both vertical and horizontal slices through the central pixel. Here, a correlation coefficient value of one signifies the pixel’s correlation with itself, acting as a reference point for the analysis.

Figure 6d,e delve into the central region of interest, along with corresponding slices. Within this region, we observe a notable long-distance correlation, characterized by a full width at half maximum of 10 pixels and a baseline of 0.4. At a 500 mm distance, this translates to an approximate 10 mm correlation. Intriguingly, the correlation coefficient consistently remains positive and does not dip below zero, highlighting that noise is predominantly correlated rather than uncorrelated.

Through this comprehensive analysis, our experimental setup and findings offer insights into the correlation of distance measurements between pixels, particularly across varying distances and within specific regions of interest. The measurements showed that the Intel RealSense L515 LiDAR has a high correlation between different pixels at long distances, e.g., 100 pixels corresponding to approximately 100 mm at a 500 mm distance. This finding might put constraints on techniques used to increase precision such as averaging over several nearby pixels. The long-distance correlation between pixels will not improve precision since distance values in nearby pixels move synchronously.

## 4. Current State and Future Direction

The proposed design embodies a modular and extensible approach to controlling software and hardware, opening up the door to a multitude of research opportunities. Presently, our control software enables the movement of the observation platform along simple trajectories. However, it is worth noting that our design inherently supports motion along complex trajectories and mounting of different augmented reality sensors. In the future, we plan to expand the use of the evaluation setup and unlock capabilities that would potentially help us to explore numerous critical questions. These include head, hand, or object tracking; static and dynamic image registration and quality; latency in head-mounted displays; temporal noise associated with moving objects or observers; and other emerging needs in sensor characterization and testing in task-based experiments.

This study aimed to evaluate the effectiveness of the proposed metrics and their application in real-world scenarios at distances typical to use of head-mounted displays in surgical settings. For example, a reduction in spatial resolution of the reported depth camera indicates that the developers should be aware that the camera might not be able to resolve to objects at a different distance that are next to each other. Looking ahead, we plan to conduct a comprehensive comparative study involving various range sensing technologies.

## Figures and Tables

**Figure 1 sensors-24-03916-f001:**
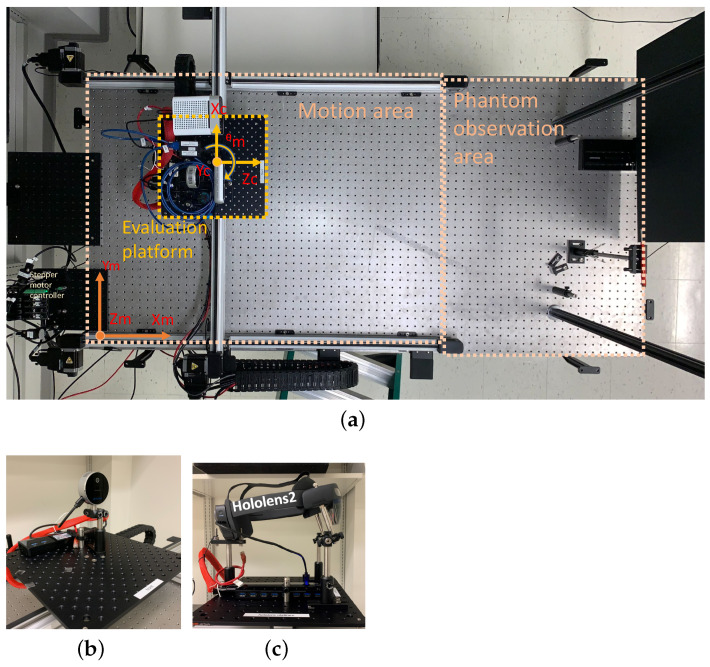
The experimental setup consists of an optical table 6 feet by 3 feet (INT3-36-8-A Integrity 3 VCS by Newport Corp). The table-top is divided into two sections: on the left in Panel A—motion control system(Zaber Technologies); and on the right in Panel A—test object observation area for positioning evaluation test objects). The motion system allows controlled motion in two horizontal axes $X_{m}$ and $Y_{m}$, and in rotational axis $\Theta_{m}$. A platform with devices under investigation can be mounted on top of the rotational stage. The depth camera generates data consisting of points in space, each assigned coordinates is noted as Xc, Yc, and Zc. Panel B shows a platform with Intel Real Sense L515 LiDAR depth camera mounted on it, and Panel C shows the Microsoft HoloLens 2 augmented reality headset mounted. (**a**) Experimental setup layout; (**b**) Depth cameras platform; (**c**) HoloLens2 platform.

**Figure 2 sensors-24-03916-f002:**
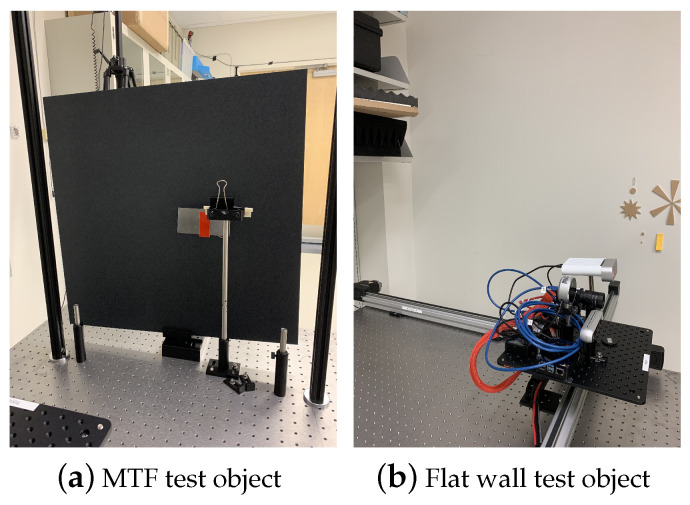
Examples of available test objects positioned in the regulatory test object observation area, from left to right: Panel (**a**)—a slanted edge test object, 60 mm tall, slanted at 4 degrees, mounted in front of a ThorLabs Black Construction Hardboard measuring 24 × 24 inches; Panel (**b**)—a segment of a flat wall test object, 2 × 2 m^2^, used for alignment of a depth camera optical axis and later measurement of a static Pearson correlation coefficient.

**Figure 3 sensors-24-03916-f003:**
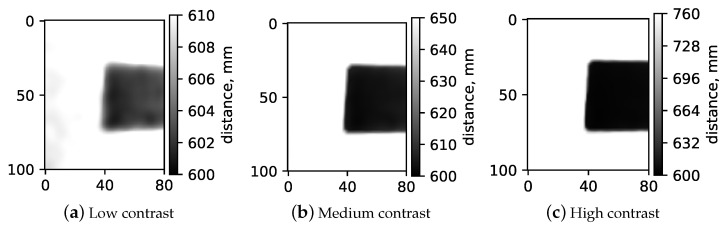
Acquired images of the slanted edge test object with varying distances between the slanted edge and the background panel, illustrating different contrast levels in depth: (**a**) low contrast at 10 mm, (**b**) medium contrast at 50 mm, and (**c**) high contrast at 160 mm. The gray scale in each figure shows the range in depth measured in the images in mm. In gray-scale heat maps, the dark area always shows the distance (600 mm) to the slanted edge since it is stationary, and the white area shows distance to the background screen. Hence, the white areas in panels (**a**), (**b**), and (**c**) represent 610 mm, 650 mm and 760 mm distances, respectively. The physical size of displayed region of interest is 100 mm (y-axis) by 80 mm (x-axis).

**Figure 4 sensors-24-03916-f004:**
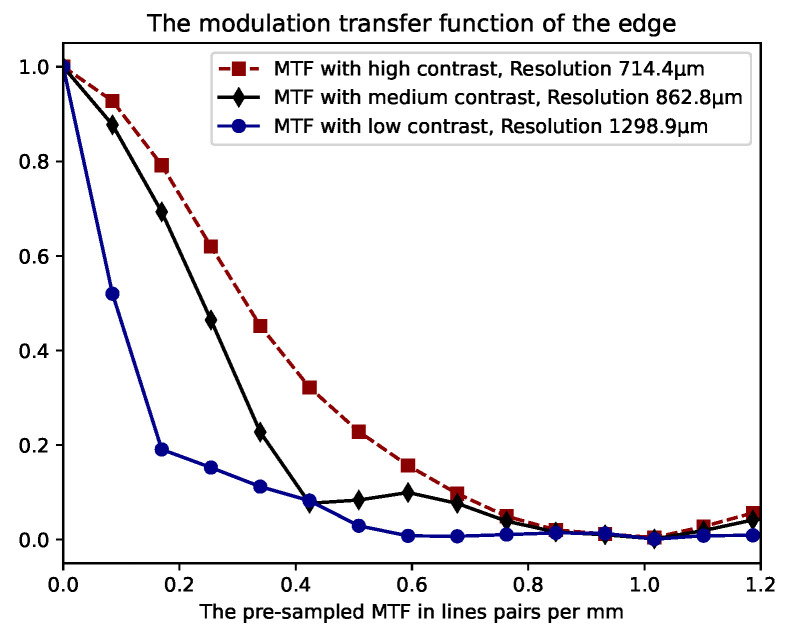
Modulation Transfer Function (MTF) calculated for three varying contrast levels, determined by the distance from the slanted edge to the background panel: low contrast at 10 mm, medium contrast at 50 mm, and high contrast at 160 mm. The resulting spatial resolutions of 714.4 μm, 862.8 μm and 1298.9 μm were for high, medium and low contrast measurements, respectively.

**Figure 5 sensors-24-03916-f005:**
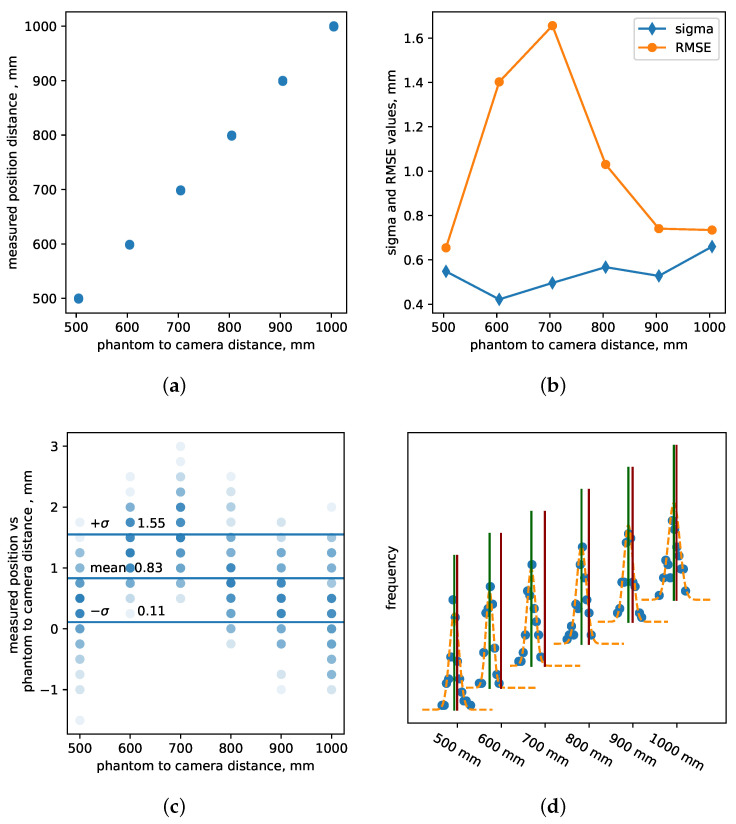
z-accuracy measurement of Intel RealSense L515 LiDAR range-sensing standalone camera. The depth camera was positioned at six fixed distances 500, 600, 700, 800, 900 and 1000 mm, and 100 images were collected at each position. Panel (**a**)—measured position of depth camera from the flat wall test object on y-axis vs. set distance by a gantry system on x-axis; Panel (**b**)—standard deviation and RMSE of every 100 images collected at each fixed camera distances; Panel (**c**)—difference between measured position and ground truth provided by motion control system, $z_{ij}-z_{j}^{truth}$, where each semi-transparant dot represents one measurement and darker color dots show overlapping measurements. The mean value and standard deviation shown on the graph corresponds to the entire dataset and ideally should be zero; Panel (**d**)—histograms of distance values measured at all distances away from the wall, where the blue circles correspond to measured data, orange show a single Gaussian fit, green vertical lines show mean value, and red vertical lines show ground truth. The difference between measured mean value and the truth shows the accuracy of the camera. The mean and standard deviation values for 500 mm distance are 499.6 mm and 0.55 mm. (**a**) Measured distance vs. test-object-to-camera distance; (**b**) Standard deviation and RMSE at each test-object-to-camera distance; (**c**) Difference in measured positions and ground truth for each camera position; (**d**) Histograms.

**Figure 6 sensors-24-03916-f006:**
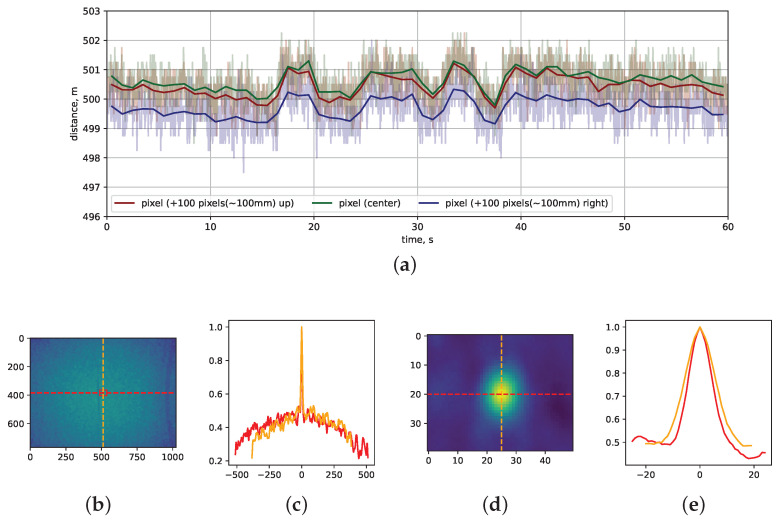
Noise measurements from Intel RealSense L515 LiDAR range sensing standalone camera and Pearson correlation analysis. Panel (**a**)—three traces of depth values measured over 60 s for three different pixels: The red solid line corresponds to a pixel 100 pixels above the central pixel, the green solid line to the central pixel, and the blue solid line to a pixel 100 pixels to the right of the central pixel; Panels (**b**,**d**) show Pearson correlation maps of central pixel vs. all other pixels with Panel (**d**) showing the central region of interest. The red square in Panel (**b**) highlights the region of interest shown in panel (**d**); Panels (**c**,**e**) show vertical and horizontal slices via central pixel, corresponding to the whole correlation map (Panel (**b**)) and the central region of interest (Panel (**d**)). (**a**) Traces; (**b**) Full-size correlation map; (**c**) Full-size slices; (**d**) Central region of interest correlation map; (**e**) Central region of interest slices.

## Data Availability

Data is available upon request.

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
