# Peer review of "Experimental Setup for Evaluating Depth Sensors in Augmented Reality Technologies Used in Medical Devices"

_sensors, 2024, doi:10.3390/s24123916_

Round 1

Reviewer 1 Report

Comments and Suggestions for Authors

1.In the abstract, it is mentioned that the device proposed in this paper is primarily intended for medical and augmented reality applications. However, in the experiments and parameter analysis of the article, there is hardly any content related to healthcare, but rather a performance analysis of camera parameters. It is not clear how these parameters (like Spatial resolution, Z-accuracy, and pixel-to-pixel correlation) are relevant and applicable in the field of healthcare.

2.In line 104, it is mentioned that, “allowing us to simulate a wide range of user activities and scenarios”. Are these activities and scenarios related with medical field,and how can they embodied in surgery or other medical activities.

3.In line 162, extrinsic alignment. It is mentioned that meticulously selected a set of five rows of pixels, can you explain specifically how these 5 lines of pixels are selected? What is the basis for selecting them, and why does vertically merging them improve the signal-to-noise ratio?

4.In line 187, “Section 3.3 looks at....... It seems to be an unfinished summary or conclusion.

5. Why is the test object in Figure 2(a) tilted, and what is the purpose of doing so?

6. The conclusions obtained in Section 3.1 do not demonstrate the relevance to the medical or augmented reality field. In another word, how does this conclusion contribute to the advancement of the augmented reality field or depth camera technology?

Author Response

Thank you for your valuable concerns and comments, which have significantly contributed to the improvement of our research paper. Please see responses to individual comments below. 

  1. In the abstract, it is mentioned that the device proposed in this paper is primarily intended for medical and augmented reality applications. However, in the experiments and parameter analysis of the article, there is hardly any content related to healthcare, but rather a performance analysis of camera parameters. It is not clear how these parameters (like Spatial resolution, Z-accuracy, and pixel-to-pixel correlation) are relevant and applicable in the field of healthcare. 

Medical extended reality devices have several constraints dictated by typical distances between a user and working area, e.g. in surgery it would be arm-length since the depth camera needs to be able to perform well while mounted on surgeon’s head in the HMD. The proposed experimental setup focuses on motion range, distance between camera and working area that correspond to typical surgical conditions. 

  1. In line 104, it is mentioned that, “allowing us to simulate a wide range of user activities and scenarios”. Are these activities and scenarios related with medical field,and how can they embodied in surgery or other medical activities. 

The text was changed to clarify what kind of scenarios authors intended to mention. Now it reads “This flexibility was essential for assessing camera responses during motion and at different locations, allowing us to simulate a wide range of user medical related activities and healthcare scenarios” 

  1. In line 162, extrinsic alignment. It is mentioned that “meticulously selected a set of five rows of pixels”, can you explain specifically how these 5 lines of pixels are selected? What is the basis for selecting them, and why does vertically merging them improve the signal-to-noise ratio? 

Binning of several rows/columns of pixels increase signal to noise making measurement more precise. Now it reads “In the horizontal ROI, we meticulously selected a set of five rows of pixels, symmetrically centered around the central row of pixel in the depth image. By vertically binning these pixels, we enhanced the signal-to-noise ratio by factor of √5 and derived a vector of distances along a horizontal axis. These central rows of pixels represent distances across the image and should be the same if the camera’s optical axis is perpendicular to the wall. This resulting vector is then subjected to a straight-line fitting process, enabling us to extract the yaw angle.” 

  1. In line 187, “Section 3.3 looks at......”. It seems to be an unfinished summary or conclusion. 

The text was added to briefly summarize what will be discussed in later sections 

  1. Why is the test object in Figure 2(a) tilted, and what is the purpose of doing so? 

One of the method for measuring resolution in optical systems is slanted edge method of measuring modulation transfer function(MTF) method. For example, https://www.strollswithmydog.com/the-slanted-edge-method/ . We utilize similar idea but we achieve contrast between dark and bright areas in terms of distance – close and far. We also added a reference to slanted edge method. 

Now reads as “The slanted edge is commonly used to determine modulation transfer function(MTF) in optical systems and allows to determine a system’s ’sharpness’ performance at all spatial frequencies in one go [17]. 

  1. The conclusions obtained in Section 3.1 do not demonstrate the relevance to the medical or augmented reality field. In another word, how does this conclusion contribute to the advancement of the augmented reality field or depth camera technology? 

The methods discussed in this work are focusing on regulatory evaluation of medical extended reality devices that use depth cameras. The abstract was modified to ensure that reader understands that the focus of this work in regulatory science and not development of augmented reality technologies. 

Now reads, 

“This study aimed to evaluate the effectiveness of the proposed metrics and their 312 
application in real-world scenarios at distances typical to use of head-mounted displays 313 
in surgical settings. For example, reduction of spatial resolution of the reported depth 314 
camera indicates that the developers should be aware that the camera might not be able to 315 
resolve to objects at different distance that are next to each other.” 

Reviewer 2 Report

Comments and Suggestions for Authors

The paper titled “Experimental Setup for Evaluating Depth Sensors in Augmented Reality Technologies Used in Medical Devices” discusses the project of a modular setup built to experimentally evaluate depth sensors, usually exploited in medical devices through Augmented Reality. The proposed system also consists of an observation platform and a test object stage. 

Major issues:

  • The authors should expand the part about the use of Augmented Reality applications done through depth devices. More examples could help the readability of the paper, and also to support the sentence “For instance, AR HMDs enable capturing of images and videos from a surgeon’s perspective” (lines 33 – 34)
  • Authors should enhance the motivations for the use of depth in HMD, in particular the aspects that will be analyzed in the following part of the paper. For instance, what is the most important feature of depth cameras in an HMD? Also in this case, these elements should help the reader understand in the Introduction section.
  • A Related Work section is missing. I believe it is important to highlight what has been done in the evaluation of depth sensors in the literature. In line 41 are reported 4 different works that can be further analyzed (i.e. [10], [11], [12] and [13]).
  • Again, in the Related Work section can be included an analysis of the use of depth data in applicative devices, such as Medical, but also Automotive (Driver Monitoring), and in different tasks (Action and Gesture Recognition, Human and Hand Pose Estimation). In all these cases, an HMD can be used: what are then the most important elements to be evaluated through the proposed platform? In other words, are “Image plane spatial resolution”, “Z-precision and Z-accuracy measurements” and “Pearson pixel-to-pixel correlation” important for these scenarios? 

Minor issues:

  • Page 5, line 187: the sentence is not complete
  • Section 3 should be expanded, including more discussion and examples.

From a general point of view, the work is well-organized and written. Unfortunately, the experimental evaluation of the platform is limited in examples and tested depth sensors. The authors should follow the aforementioned points to enhance the experimental quality of the paper, at least from a literature analysis point of view.

Author Response

Thank you for your valuable concerns and comments, which have significantly contributed to the improvement of our research paper.  

While the reviewer has provided valuable feedback, we feel that it shifts the paper's focus outside its intended scope. Our work centers on developing a regulatory science method for characterizing the spatial (X, Y) and depth (Z) resolutions of depth cameras specifically for medical extended reality applications. We aim to establish laboratory-based, reproducible measurements that can be conducted automatically, without human intervention. 

While other available methods often emphasize task-based evaluation, our proposed methods allow for the comparison of different depth sensors without the need to repeat specific tasks. This is crucial for comparing various medical devices that utilize head-mounted displays (HMDs) or depth sensors. Furthermore, our study focuses on scenarios closely related to healthcare settings, such as the medical devices that utilize HMDs in surgical environments. In these cases, there are specific constraints on distances, such as an HMD positioned at arm's length from the working area. Therefore, the discussion of automotive applications, which involve large distances between the camera and the sensing object, is not relevant to our work. 

However, we have realized that the introduction can be expanded and added few references and discussion about prior work in evaluation of depth cameras. 

Please see responses to individual comments below. 

The paper titled “Experimental Setup for Evaluating Depth Sensors in Augmented Reality Technologies Used in Medical Devices” discusses the project of a modular setup built to experimentally evaluate depth sensors, usually exploited in medical devices through Augmented Reality. The proposed system also consists of an observation platform and a test object stage.  

Major issues: 

  • The authors should expand the part about the use of Augmented Reality applications done through depth devices. More examples could help the readability of the paper, and also to support the sentence “For instance, AR HMDs enable capturing of images and videos from a surgeon’s perspective” (lines 33 – 34) 
  • Authors should enhance the motivations for the use of depth in HMD, in particular the aspects that will be analyzed in the following part of the paper. For instance, what is the most important feature of depth cameras in an HMD? Also in this case, these elements should help the reader understand in the Introduction section. 

    This work focuses on regulatory evaluation of depth sensing technologies. Hence, expanding and justifying when depth cameras are useful in augmented reality HMDs is out of scope of the paper since what is done and not why. 
  • A Related Work section is missing. I believe it is important to highlight what has been done in the evaluation of depth sensors in the literature. In line 41 are reported 4 different works that can be further analyzed (i.e. [10], [11], [12] and [13]). 

    Additional text discussing evaluation of contributions of depth sensors in few use cases has been added.  

  • Again, in the Related Work section can be included an analysis of the use of depth data in applicative devices, such as Medical, but also Automotive (Driver Monitoring), and in different tasks (Action and Gesture Recognition, Human and Hand Pose Estimation). In all these cases, an HMD can be used: what are then the most important elements to be evaluated through the proposed platform? In other words, are “Image plane spatial resolution”, “Z-precision and Z-accuracy measurements” and “Pearson pixel-to-pixel correlation” important for these scenarios?  

    The reduction of spatial resolution at low contract, e.g. hand gesture recognition very close the patient may result in inaccurate determination of the hand position leading to error in hand positioning which may result during surgical use of HMD. 

Minor issues: 

  • Page 5, line 187: the sentence is not complete 

Note and fixed 

  • Section 3 should be expanded, including more discussion and examples. 

Section 3 focuses on discussion of the example shown in the paper. We intend to follow-up with a systematic study of different depth sensing sensors using the method presented in this work. 

From a general point of view, the work is well-organized and written. Unfortunately, the experimental evaluation of the platform is limited in examples and tested depth sensors. The authors should follow the aforementioned points to enhance the experimental quality of the paper, at least from a literature analysis point of view. 

Thank you for your feedback. In this work, we demonstrate few examples of evaluation of depth sensors, and we will follow-up with systematic study of few representative depth sensors. 

Round 2

Reviewer 1 Report

Comments and Suggestions for Authors

The author has provided thorough explanations and improvements to the proposed suggestions. Now, I believe this manuscript is suitable for publication in the Sensors.

Author Response

received. Thank you.

Reviewer 2 Report

Comments and Suggestions for Authors

The authors have replied to my comment.

I suggest only adding more updated references in addition to [2] (there are more modern works about gesture recognition than in 2015).  

Comments on the Quality of English Language

No specific comments about the English language.

Author Response

Conducted literature search and added two more recent references for gesture recognition. One of which is directly related to healthcare field.